# Resilient nursing in ICU: Aadaptive practices beyond IPC protocols for MDRO management. A qualitative study

Eva Cappelli[1]*, Marianna Azzolini[2], Cristina Ferrari[2], Federica Canzan[1]

1 Department of Diagnostics and Public Health, University of Verona, Verona, Italy, 2 Health of the Professions Department, University Hospital of Verona, Verona, Italy

* eva.cappelli@univr.it

## Abstract

This descriptive qualitative study explores how resilience among intensive care nurses is activated during the management of patients with multidrug-resistant organisms and how it shapes adherence to infection prevention and control practices under conditions of clinical urgency and organisational pressure. Data were collected between March and June 2025 in two Intensive Care Units through non-participant observations and twenty-one semi-structured interviews involving nurses, a head nurse, nurse assistants, and anaesthesiologists. Interviews were audio-recorded and analysed using inductive content analysis. The analysis identified three interrelated themes encompassing eleven categories: (a) individual and team-based dimensions of nurses' resilience, (b) adaptive strategies employed by nurses in intensive care settings, and (c) dynamic interactions between professional resilience and organisational support. Resilience emerged as a multilevel and context-dependent process sustained by personal resources, teamwork, and enabling organisational conditions. These elements supported nurses in maintaining adherence to infection prevention and control protocols and safeguarding patient safety, even in highly complex and time-critical situations. However, the findings also revealed a "critical zone" of resilience characterised by constant adaptation, sustained operational pressure, and procedures perceived as difficult to implement in practice, which over time contributed to emotional fatigue and an increased risk of burnout. While professional resilience represents a crucial resource in intensive care, it cannot compensate for structural or organisational shortcomings. Sustainable infection control practices and high-quality care therefore require coherent organisational support systems that extend beyond individual and team-level resilience.

## Introduction

Healthcare-Associated Infections (HAIs) caused by Multi-Drug-Resistant Organisms (MDROs) represent a major challenge in Intensive Care Units (ICUs). The

**Data availability statement:** The data that support the findings of this study are not publicly available due to privacy and confidentiality restrictions. Requests for access to the data may be directed to the South-West Veneto Area Territorial Ethics Committee (Comitato Etico Territoriale Area Vasta Sud-Ovest Veneto), which approved this study (Prot. No. 68603). Contact information: comitatoetico@aovr.veneto.it.

**Funding:** The author(s) received no specific funding for this work.

**Competing interests:** The authors have declared that no competing interests exist.

**Abbreviations:** D, Doctors; HN, Head Nurses; HAIs, Healthcare-Associated Infections; IPC, Infection Prevention and Control; ICUs, Intensive Care Units; MRI, Magnetic Resonance Imaging; MRSA, Methicillin-Resistant Staphylococcus Aureus; MDROs, Multi-Drug-Resistant Organisms; NA, Nurse Assistants; N, Nurses; PPE, Personal Protective Equipment; SRQR, Standards for Reporting Qualitative Research; WHO, World Health Organization.

combination of clinical complexity, frequent use of invasive devices, and high patient acuity significantly increases the risk of cross-transmission [1,2]. Preventing and managing MDROs requires strict adherence to Infection Prevention and Control (IPC) protocols, whose effectiveness is influenced by multiple organisational, educational, and relational factors [3,4].

In this high-pressure context, characterised by uncertainty, heavy workloads, and at times, conflicting organisational demands, nurses play a pivotal role in mediating between formal protocols, the realities of the clinical setting, a multidisciplinary team, and the needs of critically ill patients [5,6]. This role requires advanced technical and interpersonal competencies, problem-solving abilities, and professional resilience, enabling nurses to maintain safety and care quality even under substantial organisational strain [7].

Recent theoretical perspectives on resilience in organisational contexts [8,9] highlight the interdependence of individual, team, and organisational dimensions of resilience, all of which are shaped by affective, relational, and structural dynamics [7]. Applying these concepts in ICU settings offers a valuable lens for understanding how nurses navigate everyday challenges and uphold the consistent implementation of IPC protocols in the care of patients with MDRO infections.

## Background

HAI incidence in ICUs can be up to four times higher than in other hospital wards [10], including ventilator-associated pneumonia, central-line-associated bloodstream infections, and catheter-associated urinary tract infections [4]. Beyond clinical factors and the use of invasive devices, growing evidence highlights the importance of organisational elements such as leadership, interprofessional communication, coordination, and organisational culture [2,11,12].

A major concern within ICUs is the spread of MDROs, which may account for nearly 50% of ICU-related infections [1]. Pathogens such as *Acinetobacter baumannii*, *Pseudomonas aeruginosa* and carbapenemase-producing Enterobacteriaceae pose significant challenges, often exacerbated by suboptimal adherence to IPC protocols and occasionally ineffective surveillance systems [1]. International guidelines recommend multimodal strategies including hand hygiene, the use of Personal Protective Equipment (PPE), environmental disinfection, patient screening, isolation measures, staff cohorting measures, device care bundles, continuous training, monitoring, and audits [13,14]. However, the effectiveness of these strategies cannot rely solely on staff education. Effective engagement of all medical professionals and strong managerial leadership capable of implementing structured, targeted interventions are also essential [15].

ICUs are highly complex environments. Critically ill patients require intensive care, and nurses perform more than 3,000 tasks per day, 43% of them simultaneous, creating an exceptionally high cognitive and operational load [16,17]. Such complexity demands advanced competencies, high-quality nurse–patient relationships, and effective strategies to reduce stress and prevent burnout. Prolonged exposure to critical events, staff turnover, and ethical dilemmas increase nurses' vulnerability to

emotional exhaustion, depersonalisation, and reduced professional efficacy [18,19]. These conditions compromise nurse well-being, negatively affect care quality, and heighten the risk of clinical errors [8,20].

Within this context, resilience emerges as a crucial and modifiable resource that helps nurses maintain high standards of care, manage emotional demands, and make rapid decisions in dynamic ICU environments [7]. This is particularly relevant in the management of MDRO-infected patients, where advanced skills and close team coordination are essential. Recent evidence also highlights the interdependence of individual and team resilience, which together support organisational sustainability and effective team functioning [8,9,21].

Despite this insight, the concrete mechanisms through which nursing resilience manifests in everyday ICU practice remain poorly understood. Little is known about the strategies nurses adopt when IPC protocols become difficult to apply in emergencies, or how team dynamics evolve under high pressure conditions. This study, therefore, explores how ICU nurses understand and exercise resilience during critical situations involving MDRO-infected patients. It examines the daily negotiation between IPC requirements and clinical urgency, the adaptive strategies nurses employ when protocols prove insufficient, and their influence on team dynamics, leadership, and organisational culture.

## Theoretical framework

This study draws on the broaden-and-build theory [22], which proposes that positive emotions expand individuals' cognitive and behavioural repertoires, enhancing flexibility, creativity, and the ability to manage complex or adverse situations [7]. In clinical settings, these processes support psychological well-being, reduce vulnerability to burnout, promote work engagement, and strengthen decision-making [9]. Positive emotions can also spread across the team through emotional transmission, fostering collective resilience and improving interprofessional cooperation and communication [9]. Conversely, when these dynamics are absent, nurses may experience disengagement, increased stress, and reduced team cohesion, leading them to conserve rather than invest their psychophysical resources [23].

Integrating these concepts offers insight into how nursing resilience develops through the interaction of personal resources and the work environment, while team resilience is shaped by interpersonal dynamics, mutual trust, communication quality, and leadership style [3,9]. These factors directly influence adherence to best practices, error prevention, and patient safety [7].

A qualitative approach was adopted to capture nurses' experiences and explore aspects of IPC practice that remain under-researched, particularly in the management of critically ill patients infected with MDROs. The study investigated how nurses interpret and adapt IPC protocols during patient isolation, implement necessary precautions, organise cohorting, and ensure appropriate environmental disinfection while maintaining safety for staff and other patients despite clinical and organisational constraints [14].

The analysis was framed within organisational context rather than individual experiences, following Mills' [24] concept of sociological imagination, which emphasises the interaction between personal experience and broader social structures [25]. This lens was applied to interpret the organisational challenges observed during two critical care processes: (a) the admission of a patient potentially infected with an MDRO without a definitive microbiological diagnosis, and (b) the MRI transfer of a critically ill patient infected with carbapenem-resistant *Klebsiella pneumoniae*, conducted without adequate communication between the medical–nursing team and with the radiology department.

By integrating psychological theory [22], organisational analysis [24,25], and IPC guidelines for MDRO management [4,14], this study provides a robust interdisciplinary perspective [26] that enhances understanding of how nursing resilience supports adherence to infection prevention practices in complex clinical settings. The purpose of this study is to explore how resilience among ICU nurses emerges, manifests, and is mobilised during the management of critical events involving patients infected or potentially infected with MDROs. Specifically, the study aims to examine how individual, team, and organisational dimensions of resilience influence adherence to IPC protocols under conditions of uncertainty, urgency, and organisational pressure.

## Objective

This study aimed to investigate individual and team resilience among ICU nurses, examining the role of personal resources, professional competencies, and interpersonal dynamics in caring for critically ill patients with MDRO infections. It further aimed to analyse the strategies used when IPC protocols are insufficient, assessing trade-offs between decision-making speed, patient safety, and emotional or operational load. Finally, it aimed to examine the interaction between resilience and organisational support, exploring how culture, leadership, resource availability, and structural processes shape adherence to IPC protocols and overall care quality in ICUs.

## Research questions

1. How do nurses demonstrate individual resilience when caring for critically ill ICU patients with MDRO infections? Which personal resources and professional competencies are activated?

2. What strategies do nurses employ when IPC protocols prove insufficient during critical events? How are trade-offs between decision-making speed, patient safety, and emotional/organisational load managed?

3. How do leadership, organisational culture, resource availability, and structural processes influence nursing resilience? How do these interactions support adherence to IPC protocols and overall care quality in ICUs?

## Materials and methods

### Design

A descriptive qualitative approach was adopted to capture participants' perspectives on everyday challenges in managing IPC protocols in ICUs [27], allowing their narratives to remain closely aligned with their own language and lived experiences, offering a clear, realistic, and contextually grounded account of what was observed in practice [28].

This methodology enabled in-depth exploration of nurses' experiences, including those that are implicit or difficult to quantify, and how these intersected with care practices, IPC adherence, and team collaboration, allowing for a rich understanding of ICU operational processes.

The study was conducted and reported in accordance with the Standards for Reporting Qualitative Research (SRQR) [29] (see S1File).

### Study setting and sampling

The study was conducted in two ICUs (cardiac surgery and neurosurgery) within a university hospital in northern Italy with advanced infection control and antimicrobial stewardship programmes in place.

The cardiac surgery ICU contained 18 beds, including four single rooms primarily used for heart transplant recipients and for patients infected with MRSA following cardiac surgery. The neurosurgical ICU included 12 beds, also with four single rooms, occasionally used for organ donor patients requiring specific isolation conditions.

A purposive snowball sampling approach [30] was used to recruit nurses, support staff, head nurses, and anaesthetists with at least six months of ICU experience.

### Inclusion and exclusion criteria

Inclusion criteria ensured that participants were sufficiently familiar with organisational dynamics and able to articulate how nurses maintained IPC adherence under external pressures. Selection criteria included: (a) at least six months' ICU experience, and (b) representation across a range of ages and professional seniority to reduce bias and enhance transferability and reliability.

Participation was voluntary. Eligible staff were contacted via institutional email following approval from nursing management. Before enrolment, participants received detailed information regarding study aims, procedures, and confidentiality. Recruitment continued until data saturation was reached, defined as the point at which no new significant insights emerged [31].

## Instrument with validity and reliability

Rigour was ensured using strategies consistent with established qualitative reliability criteria [32]. Credibility was supported by the involvement of a multidisciplinary research team experienced in qualitative methods and nursing practice. Interviews were conducted by a researcher with expertise in qualitative methods and ICU infection risk management, and the data were independently analysed.

Coding processes were discussed and compared within the research team to support confirmability and analytical reliability [33]. Transferability was enhanced through purposive sampling and detailed contextual description, enabling readers to assess applicability to similar settings.

## Data collection

Data collection was conducted between 1 March and 30 June 2025 through non-participant observations and semi-structured interviews, which were informed by WHO guidelines [14] and hospital IPC procedures.

**Phase 1: Non-participant observations.** Non-participant observations focused on staff behaviours and interactions (Table 1), offering an initial understanding of organisational context and care processes related to managing MDRO-infected patients [4,34].

Observations targeted two specific care processes. The sequences of events observed were documented as narrative case studies and used during the semi-structured interviews as contextual reference tools to facilitate discussion with healthcare professionals about the organisational challenges identified. For the purposes of this manuscript, these case studies have been transformed into two observational vignettes, Fig 1 and 2 [35] that synthesise the key organisational challenges encountered. Observations were conducted by experienced ICU nurses external to the hospital to ensure independence. Each ward was observed for an average of three hours across different shifts (morning/afternoon), totalling 30 hours.

**Table 1. Guide for Non-Participant Observations.**

| Action | Healthcare Professional Involved | Adherence To Best Practices |
|---|---|---|
| Use of standard precautions in the management of HAIs from MDROs | Physician, Nurse, Head Nurse, Nurse Assistant, External Consultants, or others | Yes/No/Other |
| Use of standard precautions in the management of device-related HAIs from MDROs | Physician, Nurse, Nurse Assistant | Yes/No/Other |
| Use of staff cohorting for patients with an HAI caused by an MDRO | Physician, Nurse, Nurse, Head Nurse, Nurse Assistant | Yes/No/Other |
| Management of intra-hospital transport of a patient infected with an HAI from an MDRO | Physician, Nurse, Head Nurse, Nurse Assistant | Yes/No/Other |
| Management of environmental sanitation | Nurse, Head Nurse, Nurse Assistant | Yes/No/Other |

Structured observational guide used to assess adherence to IPC best practices among ICU healthcare professionals in the management of HAIs caused by MDROs. (HAI: Healthcare-Associated Infection; MDRO: MultiDrug-Resistant Organism; IPC: Infection Prevention and Control; ICU: Intensive Care Unit).

 

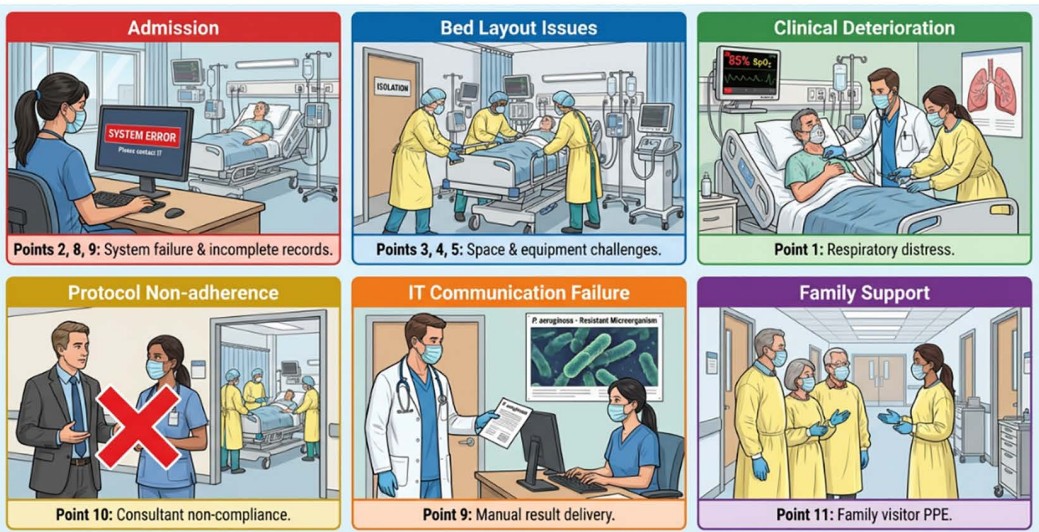

**Fig 1. Observational vignette 1-ICU admission in the context of suspected MDRO infection.** Illustrative storyboard synthesising the key organisational challenges (Points 1–11) observed by the researcher during non-participant observation sessions in the Intensive Care Unit. The vignette was used as a contextual reference tool to facilitate semi-structured interviews with healthcare professionals, enabling participants to articulate their experiences around IPC adherence, workload management, and interprofessional communication without requiring disclosure of personal information. The visual format was produced using an AI-assisted illustration tool (SciSpace; https://scispace.com) on the basis of a structured textual description provided by the research team. No individual patient data are represented or reported. ICU: Intensive Care Unit; MDRO: Multi-Drug-Resistant Organism; IPC: Infection Prevention and Control; PPE: Personal Protective Equipment.

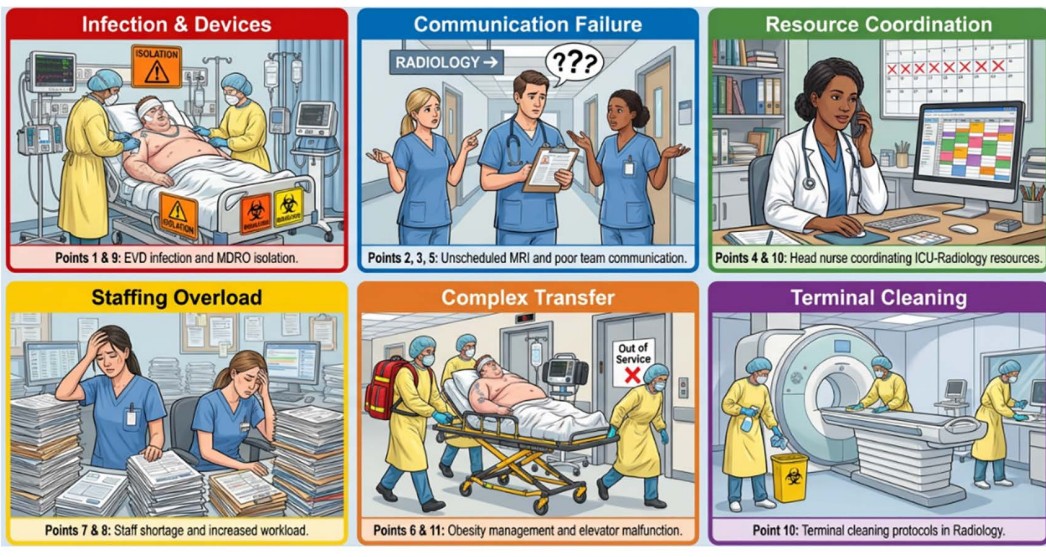

**Fig 2. Observational vignette 2- Intra-hospital transfer of an MDRO patient to Radiology.** Illustrative storyboard synthesising the key organisational challenges (Points 1–11) observed by the researcher during non-participant observation sessions in the Intensive Care Unit. The vignette was used as a contextual reference tool to facilitate semi-structured interviews with healthcare professionals, enabling participants to articulate their experiences around IPC adherence, workload management, and interprofessional communication without requiring disclosure of personal information. The visual format was produced using an AI-assisted illustration tool (SciSpace; https://scispace.com) on the basis of a structured textual description provided by the research team. No individual patient data are represented or reported... ICU: Intensive Care Unit; MDRO: Multi-Drug-Resistant Organism; IPC: Infection Prevention and Control; MRI: Magnetic Resonance Imaging; PPE: Personal Protective Equipment.

**Phase 2: Semi-structured interviews.** Semi-structured interviews were conducted following a predefined interview protocol [34] and explored the strategies used by nurses to maintain IPC adherence despite clinical urgency or organisational constraints. Sociodemographic and professional data (sex, age, education, years of experience, and unit affiliation) were also collected.

Interviews were conducted individually in a private and comfortable setting and scheduled around participants' shifts. Each interview lasted 40–90 minutes, was audio-recorded with consent, and fully transcribed for analysis. Two external researchers conducted all interviews to enhance impartiality and reduce contextual bias.

## Data analysis

Data were analysed using qualitative content analysis, a systematic method for coding and categorising text to identify recurring patterns and themes [36]. Transcripts were independently read multiple times for familiarisation.

An inductive process was used to identify meaningful units of text and to develop initial codes reflecting emerging ideas. Codes were refined to reduce redundancy, grouped into broader categories, and then synthesised into subthemes based on frequency, structural similarity, and logical relationships. Subthemes were collaboratively reviewed to ensure that they aligned with the study's objectives.

## Ethical considerations

The study adhered to the Declaration of Helsinki [37]. Ethical approval was obtained from the Territorial Ethics Committee – South-West Veneto Area (IPC-coordinf – Prog. 422CET). All participants received full information about the study and provided written informed consent prior to participation. Anonymity was ensured through the use of unique identifiers. All personal data were removed from transcripts, and audio files were securely stored on a secure institutional server.

## Results

### Characteristics of the sample

In total, 21 participants were interviewed, including nine Nurses (N), two Head Nurses (HN), four Nurse Assistants (NA), and six Doctors (D). Participants ranged in age from 26 to 60 years, and 70% were women. Overall professional experience varied from 1 to 35 years, while tenure within the same ICU ranged from 1 to 30 years (Table 2).

Thematic analysis identified three main themes, further divided into eleven categories (Table 3), aligned with the study's aim to explore the role of nursing resilience in managing critically ill patients infected with MDROs and in adhering to IPC protocols.

### General overview of results

Overall, the findings indicated that resilience cannot be understood solely as an individual attribute. Rather, it develops through the interplay of personal resources, professional collaboration, and organisational conditions. The three themes reflect three distinct but interconnected levels of resilience: (a) individual and team-based dimensions of nurses' resilience,

**Table 2. Participants' characteristics.**

| Variable | Nurse (9) | Doctor 96) | Nurse Assistant (4) | Head Nurse (2) |
|---|---|---|---|---|
| *Sex* | | | | |
| *Female* | 56% | 50% | 100% | 100% |
| *Male* | 44% | 50% | – | – |
| *Age (average) SD* | 36 (10.5) | 43.7 (8.3) | 51.5 (8.27) | 53 (1.41) |
| *Work experience (average) SD* | 11.1 (8.3) | 14 (9.1) | 18.25 (6.48) | 32 (4.24) |
| *Work experience in ICU (average) SD* | 8.1 (7.5) | 8 (7.2) | 6 (2) | 14,5 (6.36) |

*Sociodemographic and professional characteristics of study participants (n = 21), including sex distribution, mean age, and work experience by professional role. (SD: Standard Deviation; ICU: Intensive Care Unit).*

**Table 3. Themes, categories, and codes emerging from inductive content analysis of semi-structured interviews with intensive care unit staff.**

| Theme | Category | Code |
|---|---|---|
| **Theme 1 Individual and Team Resilience of Nurses** | *Self-Efficacy: Personal Efficacy and Situational Initiative* | *Individual initiative; situational adaptability; operational readiness; emergency management; mutual support.* |
| | *Professional Competencies: advanced technical skills* | *Supervision and mentoring; management of complex devices; accurate documentation; care planning; insufficient integrated training; continuous updating needs.* |
| | *Positive Emotions: emotional stability and regulation* | *Stress management; empathic support; team trust.* |
| | *Nurse Well-being: stress factors* | *Organizational stressors; psychological factors; relational factors; team dynamics impact.* |
| **Theme 2 Nurse Adaptive Strategies in Intensive Care Unit** | *Dynamic adaptation of spaces and protocols* | *Limited space optimization; PPE implementation and management; procedure updates and adaptation; resilience under pressure.* |
| | *Managing trade-offs between rapid decisions and safety* | *Critical decision making; risk prevention strategies.* |
| | *Creative and proactive problem-solving* | *Innovative solutions development; resource optimization; infectious waste management.* |
| | *Cooperative communication and coordination* | *Interdepartmental coordination; peer monitoring and support; clinical documentation; structured briefing needs.* |
| **Theme 3 Interaction between Nurse Resilience and Organisational Support** | *Nursing leadership* | *Supervision and mentoring; strategic delegation.* |
| | *Available resources and structured IPC processes* | *Adaptation to constraints; infrastructure limitations; communication gaps.* |
| | *Organisational culture of infection risk* | *Adherence to procedures; heterogeneity of protocols; continuous monitoring; communication delays; shared IPC culture development.* |

(b) adaptive strategies employed by nurses in intensive care settings, and (c) dynamic interactions between nurse resilience and organisational support. This interdependence directly influenced patient safety and the effectiveness of MDRO infection prevention and control practices (Fig 3). Alongside protective factors, a less discussed dimension of resilience also emerged: a "dark zone" marked by forced adaptation, frustration, overload, and risk of burnout; reflecting processes described by Hobfoll's [23] conservation of resources theory.

### Theme 1: Individual and team resilience of nurses

The first theme described how ICU nurses mobilised personal resources, professional competencies, positive emotions, and interpersonal dynamics [8,38] to ensure patient safety and continuity of care when managing critically ill patients infected with MDROs. Consistent with Hartmann et al.'s framework [9], these individual and team-level resources represent the foundational layer of resilience, upon which adaptive strategies and organisational responses are built.

**1.1 Self-Efficacy: personal efficacy and situational initiative.** Nurses demonstrated high levels of self-efficacy and initiative in enforcing isolation measures, including with staff external to the ward. In several instances, prompt implementation and reinforcement of IPC protocols proved decisive, as recounted by a head nurse:

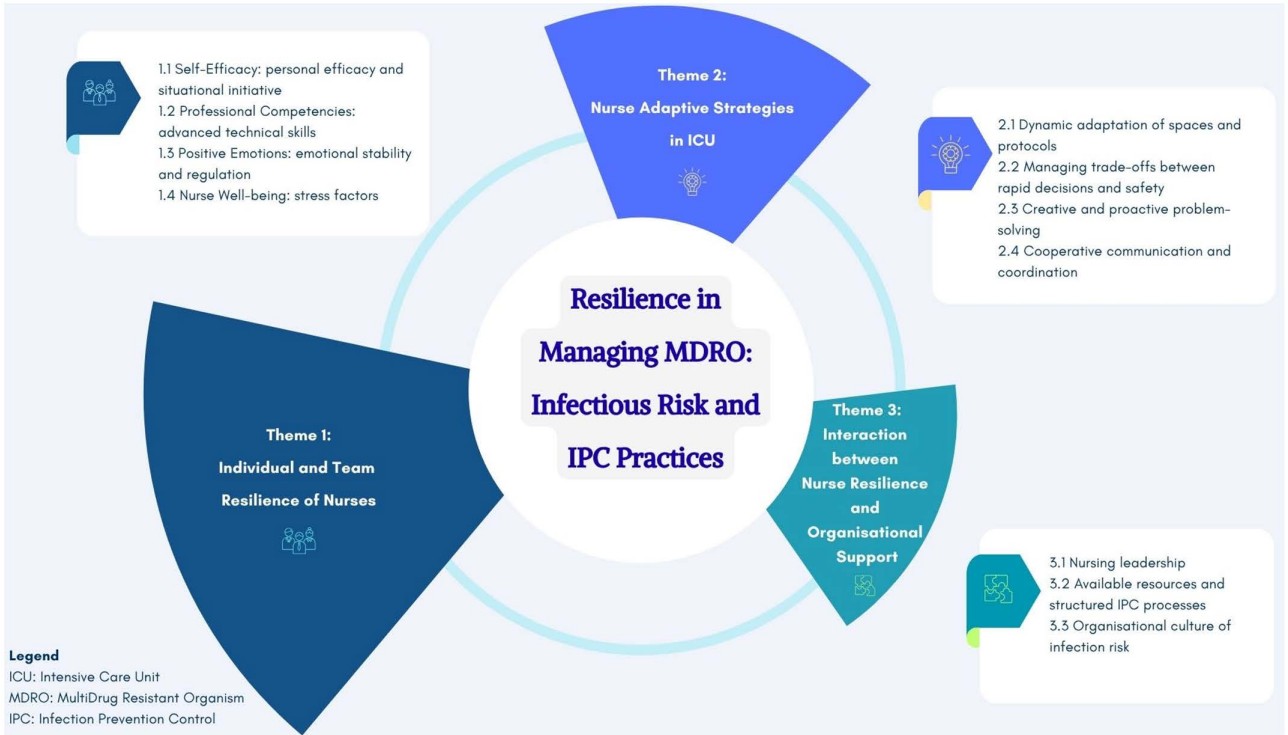

**Fig 3. Thematic overview of resilience processes and adaptive strategies in ICU nursing for MDRO management.** The figure illustrates three interconnected themes, individual and team resilience of nurses, nurse adaptive strategies in ICU, and the interaction between nurse resilience and organisational support, converging around the central construct of resilience in managing infectious risk and IPC practices. Each theme is detailed by its constituent categories, highlighting the multilevel and dynamic nature of resilience in high-complexity clinical settings. ICU: Intensive Care Unit; MDRO: MultiDrug-Resistant Organism; IPC: Infection Prevention and Control.

*"A newly hired nurse caring for a patient reminded the cardiac surgeon and their trainees to comply with PPE use to respect isolation rules for the infected patient"* [HN2].

Similar assertiveness was reported across other high-complexity situations, including intra-hospital patient transfers (Table 4).

**1.2 Professional competencies: advanced technical skills.** Resilience also manifested through advanced skills in organisational space, managing invasive devices, and planning workloads. Nurses described adapting the care environment under suboptimal conditions:

*"Available spaces are often limited and cluttered… with the support nurse, we tried to redesign and keep the passage clear"* [N1].

Device management was equally central, with nurses maintaining rigorous IPC practices even for complex procedures:

*"We ensured the sterile management of the external ventricular drain… using PPE each time before approaching the patient"* [N4].

Workload reorganisation and real-time documentation were also identified as key competencies supporting IPC adherence (Table 4).

**Table 4. The most representative quotes by category and theme discussed in the results.**

| Theme | Category | Participant | Code |
|---|---|---|---|
| Theme 1. Individual and Team Resilience of Nurses | Self-Efficacy: personal efficacy and situational initiative | N3 | *"I managed a patient transfer to the MRI; it was very challenging with many devices, monitors, and limited space… I tried to ensure the best adherence to PPE use".* |
| Theme 1. Individual and Team Resilience of Nurses | Professional Competencies: advanced technical skills | HN1 | *"I reorganised the nurse's workload, temporarily assigning another patient to a nearby colleague".* |
| Theme 1. Individual and Team Resilience of Nurses | Professional Competencies: advanced technical skills | N3 | *"We always update all device-maintenance information promptly, which is often more challenging in the ICU than in other wards".* |
| Theme 1. Individual and Team Resilience of Nurses | Nurse Well-being: stress factors | N3 | *"Excessive administrative burdens and exhausting shifts"* |
| Theme 2. Nurse Adaptive Strategies in the ICU | Dynamic adaptation of spaces and protocols | N7 | *"We used screens to create a 'bubble' reminding staff to wear PPE before entering".* |
| Theme 2. Nurse Adaptive Strategies in the ICU | Dynamic adaptation of spaces and protocols | N3 | *"Hand sanitiser dispensers were placed strategically… near the service table and monitors".* |
| Theme 2. Nurse Adaptive Strategies in the ICU | Managing trade-offs between rapid decisions and safety | N9 | *"One of us stays outside the room to pass all necessary items without risking contamination".* |
| Theme 2. Nurse Adaptive Strategies in the ICU | Cooperative communication and coordination | N5 | *"Or I remind my distracted colleague who forgot to wear the gown correctly".* |
| Theme 2. Nurse Adaptive Strategies in the ICU | Cooperative communication and coordination | N4 | *"Promptly calling Radiology to confirm the MRI booking for an obese, infected patient".* |
| Theme 3. Interaction between Nurse Resilience and Organisational Support | Available resources and structured IPC processes | N3 | *"Poorly functioning lifts designated for infected patients".* |
| Theme 3. Interaction between Nurse Resilience and Organisational Support | Available resources and structured IPC processes | N3 | *"Gaps in interdepartmental communication and incomplete paper documentation".* |
| Theme 3. Interaction between Nurse Resilience and Organisational Support | Organisational culture of infection risk | N1 | *"Despite clear signs and instructions".* |

Representative quotes by theme and category illustrating nurses' resilience, adaptive strategies, and organisational dynamics in the management of MDRO-related HAIs in the ICU. (HAI: Healthcare-Associated Infection; MDRO: MultiDrug-Resistant Organism; IPC: Infection Prevention and Control; ICU: Intensive Care Unit; N: Nurse; HN: Head Nurse).

**1.3 Positive Emotions: emotional stability and regulation.** Nurses' emotional stability supported decision-making under high clinical intensity.

*"We [remained] calm and controlled despite the complexity of patient care and physician pressure, even with unexpected transfers and logistical challenges with an obese, infected patient"* [N6].

These accounts reflect the principles of the *broaden-and-build theory* [22] whereby positive emotional states expand cognitive and behavioural repertoires, enabling nurses to sustain effective decision-making even under intense clinical pressure.

The team also demonstrated empathy and sensitivity towards the emotional needs of family members, delegating specific welcoming tasks to support staff.

*"Family members are overwhelmed by grief and despair… You must instruct them on hand sanitisation before and after visiting their loved one, or how to wear gowns and masks correctly to prevent contamination"* [NA1].

**1.4 Nurse well-being: stress factors.** Participants reported managing multiple critically ill patients simultaneously, maintaining *"constant hyper-vigilance"* [N3], and being frequently exposed to *"infection risk, including the possibility of*

*transmitting pathogens to their own families" [N5]*. These stressors contributed to cognitive overload and frustration. Such multidimensional pressures are well-established predictors of burnout and moral distress in ICU settings [20], and the narratives collected here illustrate how their accumulation can erode individual resilience over time. Episodes of demotivation and professional isolation were also linked to perceived lack of recognition:

*"I am neither recognised nor respected by some ward physicians and external consultants" [N4].*

Chronic staffing shortages further compounded these pressures, increasing the risk of burnout (Table 4).

### Theme 2: Nurse adaptive strategies in the ICU

The second theme highlighted how nurses developed adaptive strategies to manage uncertainty, structural limitations, and organisational pressures when caring for MDRO-infected patients. These strategies stemmed from individual experience and professional training [6,39].

**2.1 Dynamic adaptation of spaces and protocols.** A key adaptive skill involved reshaping IPC protocols and physical spaces under suboptimal conditions, implementing temporary yet safe solutions. Nurses described demarcating areas and restricting materials to contain contamination risk:

*"We demarcated the area… limited the materials… to prevent contamination" [N2].*

Additional strategies included the use of physical barriers and strategic placement of PPE dispensers to create functional isolation zones even in open-space settings (Table 4).

**2.2 Managing trade-offs between rapid decisions and safety.** During an acute episode of respiratory distress followed by cardiac arrest in a patient infected with carbapenem-resistant *Klebsiella pneumoniae*, physicians and nurses rapidly recalculated procedures and timings to ensure survival.

*"When patient monitor alarms sound, multiple life-saving manoeuvres are performed urgently; one must balance speed and safety, and precautions are not always fully applied. The patient must be saved immediately. After the emergency, we return to order and follow protocols carefully. We change uniforms after the emergency to avoid external contamination" [D4].*

This account illustrates the tension between protocol adherence and clinical urgency that characterises high-acuity settings, requiring nurses to exercise rapid judgement and situational prioritisation, core components of adaptive resilience under pressure [6,39].

In less urgent situations, the team developed organisational workarounds to maintain safety without interrupting care (Table 4).

**2.3 Creative and proactive problem-solving.** Nurses also optimised materials, replaced reusable devices with disposable ones, redefined routes, used colour-coded signage, involved support staff, performed microbiological sampling, and monitored environmental sanitisation.

*"When a critical patient's ventilator malfunctioned, we immediately replaced the equipment and disposable tubing, coordinating with the technician. Meanwhile, two colleagues ensured care for other patients and verified proper sanitisation of the faulty ventilator, preventing any contamination risk" [N5].*

**2.4 Cooperative communication and coordination.** Team resilience was evident in peer supervision and support for newly hired staff:

*"We support new staff facing unwelcoming dynamics from long-standing physicians"* [N4].

The head nurse played a key role in ensuring information flow:

*"She contacted the Emergency Department head nurse to understand why handovers were missing"* [N3].

The need for more structured interprofessional communication was also highlighted:

*"We need more structured briefings with physicians to manage patients and critical events… often our activities are automatic rather than deliberate choices"* [N2].

These findings underscore the relational dimension of team resilience, highlighting how structured interprofessional communication serves not only as an operational tool but as a mechanism for sustaining collective adaptive capacity.

**Theme 3: Interaction between nurse resilience and organisational support**

The third theme illustrated how leadership, structural resources, and organisational culture significantly impacted IPC protocol compliance in the ICU.

**3.1 Nursing leadership.** Beyond formal supervision by the head nurse, leadership at the bedside was also expressed through initiative and the ability to implement decisions:

*"I personally checked the line, delegated ventilator management to the anaesthetist, and informed the coordinator… despite neurological urgency, we maintained IPC procedures"* [N5].

**3.2 Available resources and structured IPC processes.** Nurses adapted protocols to operational conditions, acknowledging that structural constraints required continuous re-engineering of procedures:

*"The substance doesn't change, but the approach does… when unpredictable, re-engineering is necessary"* [N2].

Persistent infrastructural deficiencies, including malfunctioning dedicated equipment and fragmented interdepartmental communication, are reported in Table 4 and were further illustrated in the observational vignettes, Fig 1 and 2.

**3.3 Organisational culture of infection risk.** Variability in IPC adherence was observed among external consultants, reflecting uneven training and awareness across professional groups. Participants highlighted systemic discontinuities between care settings:

*"The same type of patient is managed differently in the ICU and rehabilitation"* [N2].

Nurses actively promoted a culture of infection safety, often acting as *"guardians of the rules"* [D2], a role that, while essential, generated additional pressure beyond their primary caregiving responsibilities. The head nurse further articulated the broader organisational challenge:

*"Heterogeneous protocols between hospitals and rehabilitation facilities, incomplete or delayed communications, for example, delays in reporting MDRO cases between laboratories, wards, and Radiology"* [HN1].

Such discontinuities hindered the development of a genuinely shared IPC culture. From a theoretical perspective, these findings align with Hartmann et al.'s [9] conceptualisation of organisational resilience as dependent on coherent

institutional structures and shared professional norms, conditions that, when absent, place a disproportionate burden on individual and team-level adaptive resources.

## Discussion

Drawing on the multi-level conceptualisation proposed by Hartmann et al [9] the findings of this study showed that ICU nurses' resilience becomes especially visible when caring for patients with MDROs under conditions of extreme instability. In these scenarios, nurses had to continuously negotiate between urgent clinical priorities and strict IPC requirements, often in situations characterised by spatial limitations, insufficient resources, and sudden critical deterioration, illustrating how resilience unfolds across individual, team, and organisational levels (Fig. 3). Nurses' capacity to manage stress, make rapid decisions, and cope with emotional pressure depended not only on advanced clinical competencies [6,39] but also on effective interprofessional relationships, communication, and the structural support available within the work environment.

### Individual resilience, team dynamics, and leadership in the management of patients with MDROs

Participants described resilience as a combination of self-efficacy, advanced clinical skills, emotional regulation, and proactive engagement in complex contexts. These findings, detailed in Theme 1, show that nurses maintained composure, reorganised tasks, and supported colleagues even in the most demanding clinical situations. These elements reflect the principles of the *broaden-and-build theory* [22] which argues that positive emotions expand cognitive and behavioural repertoires, thereby supporting flexibility, creativity, and rapid decision-making in high-pressure situations [7]. In practical terms, fostering emotional regulation and self-efficacy in ICU nurses may represent a key target for resilience-building interventions in clinical practice. These findings have direct implications for ICU nurse recruitment and onboarding. Organisations should invest in structured induction programmes that explicitly develop self-efficacy, emotional regulation, and assertiveness in newly hired staff, rather than relying on informal socialisation processes that may perpetuate hierarchical imbalances.

In the highly pressurised ICU environment, these processes also occurred at the team level, influencing collective attitudes and behaviours and promoting trust, cohesion, and psychological safety, although at times these were hindered [8,9]. Mutual support, such as *"timely sharing of information, assistance with invasive devices, or redistribution of tasks when a colleague was overloaded" [N6]* (Theme 2)*,* enabled teams to manage complex scenarios without individual isolation or overload. Supervision and mentoring further facilitated adherence to IPC among newer staff members, illustrating how individual and team resilience are interdependent [8,9,21]. At the team level, these results suggest that structured interprofessional briefings and debriefing sessions, particularly after critical events, could formalise the informal coordination mechanisms observed in this study, reducing cognitive load and preventing the accumulation of unaddressed tensions [16,17]. The interaction between individual resilience, team cooperation, and strong clinical leadership was found to form the basis of organisational resilience [9,11], particularly evident in the management of clinically unstable MDRO patients, where nurses needed to balance rapidly changing clinical conditions with IPC requirements. Without a supportive organisational culture, the effectiveness of resilience was reduced, heightening both emotional and operational pressures [3,12].

### Nurse well-being, stressors, and advanced clinical practice

Findings from Theme 1 (category 1.4) showed that nurse well-being emerged as a central component in sustaining resilience and clinical safety [7]. Participants described numerous, multidimensional stressors, including decision-making under uncertainty, urgent clinical situations, simultaneous management of complex protocols, persistent structural deficiencies, delayed communication about potentially infected patients, limited risk awareness among consultants, and high

responsibility for invasive devices and medication management. These factors align with established predictors of burnout and moral distress [20], which are an inherent part of everyday ICU work [6].

Nurses responded with resilient behaviours characterised by initiative, coordination, and emotional stability. Advanced clinical skills, anticipation of problems, autonomous management of non-conformities, interdisciplinary coordination, situational mentoring, and patient- and context-specific adaptation of IPC protocols helped maintain a *"sense of control and mastery" [N3]*, essential for psychological well-being [17].

These findings suggest that supporting nurses' advanced clinical practice and role recognition may be essential to sustaining resilient behaviours over time. However, several narratives revealed the "dark side" of resilience: the perceived need to endure at all costs, pressure to hide fatigue and vulnerability, difficulties seeking support, feelings of isolation and demotivation due to limited role recognition, and persistent fear of errors or contamination. Nurses also expressed frustration at inconsistent practices in other units or by other professionals, which sometimes reduced their willingness to maintain resilient behaviours. These accounts illustrate what the literature has termed "forced" or "dark side" resilience [23] a condition in which the sustained effort to maintain performance conceals underlying distress and increases vulnerability to burnout. This has direct implications for practice: organisations must move beyond expecting resilience as an individual capacity and instead create structural conditions that prevent forced adaptation.

It is important to acknowledge a critical limitation of resilience as both a theoretical and practical framework. When resilience becomes an institutional expectation rather than a personal resource, it risks functioning as a mechanism through which systemic failures are displaced onto individual practitioners, normalising structural deficiencies by valorising adaptive capacity as a professional virtue. Drawing on Hobfoll's conservation of resources theory [23], this process can be understood as a form of resource depletion: when nurses are repeatedly required to compensate for organisational shortcomings, their personal and professional resources are progressively exhausted, increasing vulnerability to burnout and moral distress. The present findings provide empirical support for this concern, as several participants described feeling obligated to *"always endure",* suggesting that resilience had shifted from a protective resource to an additional occupational burden.

Several ICU-specific barriers to sustainable resilience were identified in this study, including chronic understaffing, inadequate isolation infrastructure, fragmented inter-departmental communication, and inconsistent IPC training across professional groups (Theme 3, categories 3.2 and 3.3). These barriers are consistent with those reported in recent literature on ICU nursing complexity and workload [6,16,17], on the organisational determinants of HAI prevention [2,11,12], and on the relationship between nurse staffing, turnover, and care quality [19]. Taken together, these findings suggest that resilience-based interventions, in the absence of concurrent structural investment, are unlikely to produce lasting improvements in safety or staff well-being.

These findings both align with and extend existing evidence. Consistent with Hassan and Elsayed [7], nurses in this study demonstrated continuous adaptation, collaborative unity, and emotional balance as core resilience mechanisms. However, unlike studies conducted in less acute or less infection-intensive settings, the present research highlights a tension distinctive to ICU environments: the simultaneous management of life-threatening clinical instability and strict infection control requirements amplifies both the demands placed on resilience and the consequences of its failure [1,2,4]. This dual pressure, clinical and infectious, represents a specific burden not fully captured in existing resilience frameworks [8,9] and warrants targeted attention in future research and organisational policy [15].

## Balancing IPC protocols and clinical urgency: resilient strategies in ICUs

A persistent tension emerged between IPC protocols and clinical urgency, where rapid decision-making and immediate interventions often made strict adherence to guidelines, usually designed for ideal conditions, challenging. As illustrated in Theme 2 (categories 2.1 and 2.2), nurses described situations in which the immediate need to save a patient's life required temporary deviation from standard IPC sequences, followed by careful restoration of protocols once the

emergency had passed. Protocols assume precise operational sequences, immediate resource availability, timely communication, and adequate isolation spaces and pathways [4,14]. Yet these conditions were frequently unmet because of structural constraints, inefficient logistics, incomplete or delayed information, sudden critical deterioration, and staffing shortage [18,19]. In response, nurses developed situation-specific adaptive strategies that went beyond procedural adjustments [3].

These strategies, documented across Theme 2 and Theme 3, included rapid micro-decisions to limit exposure, reorganisation of care pathways, temporary demarcation or repurposing of spaces, immediate sanitisation, mutual supervision, and constant negotiation with external services. Such adaptations enabled acceptable safety levels even under suboptimal conditions, demonstrating that MDRO management requires ongoing reinterpretation and optimisation of protocols through resilient skills, initiative, coordination, and creative problem-solving.

Without a supportive organisational culture, these adaptive strategies become more demanding, affecting psychological well-being and the sustainability of resilient practices. Nonetheless, the need for adaptation also highlighted resilience's transformative potential: integrating technical skills, situational leadership, and team collaboration to enhance patient safety, foster operational innovation, and improve organisational efficiency [6,39]. These findings suggest that IPC training programmes should incorporate resilience-based approaches, preparing nurses not only to follow protocols but to safely navigate situations where strict adherence is temporarily compromised. At the organisational level, the findings point to the urgent need for integrated communication systems, adequate nurse-to-patient ratios, and clear protocols for intra-hospital patient transport involving MDRO-infected patients. These are not merely logistical improvements but prerequisites for sustaining safe and resilient nursing practice. In the absence of such support, resilience is unlikely to be sustained, leading to negative consequences for staff well-being and quality of care [23].

The findings of this study carry concrete practical implications for three interconnected areas of ICU organisation. First, regarding staffing: adequate nurse-to-patient ratios are a prerequisite, not a luxury, for resilient ICU practice. The data demonstrate that chronic understaffing does not merely increase workload [16,17] it fundamentally undermines the conditions under which individual and team resilience can develop and be sustained, with direct consequences for IPC adherence and MDRO containment [1,2]. Healthcare organisations should therefore treat staffing levels as a patient safety variable rather than a purely administrative one [19]. Second, regarding mentoring: the findings highlight the pivotal role of experienced nurses and head nurses in supporting newly hired staff through hierarchically complex and infection-sensitive situations [11]. Formalising mentoring relationships, particularly during the first months of ICU practice, could systematically transfer the tacit resilience competencies observed in this study, reducing both clinical risk and the emotional isolation reported by junior nurses [8,9]. Third, regarding mental health support: the "forced resilience" dynamic identified in this study, characterised by pressure to conceal fatigue, minimise vulnerability, and maintain performance at personal cost, represents a significant and under-addressed occupational health risk [20,23]. Healthcare organisations should implement proactive, non-stigmatising psychological support programmes for ICU staff, including structured peer support, access to clinical psychology services, and regular well-being monitoring, as integral components of any organisational resilience strategy [7,18].

## Strengths and limitations

This study presents several strengths. Conducting the research in real clinical settings, specialised ICUs caring for patients with MDROs daily, enhanced the practical relevance of the findings. The participation of professionals with diverse backgrounds and experience levels enabled the integration of multiple perspectives, providing a holistic understanding of the issues explored. Triangulation through interviews, observations, and organisational documents enabled an in-depth examination of individual and team resilience, adaptive strategies, and nursing leadership, resulting in a rich and detailed picture of everyday practices.

However, several limitations must be acknowledged. The study was conducted in only two ICUs, which restricts transferability to other hospital contexts. The use of purposive and snowball sampling, while appropriate for this qualitative design, introduces an inherent risk of selection bias; the snowball mechanism may have disproportionately engaged nurses with higher professional visibility, potentially excluding perspectives associated with disengagement or critical views of organisational practices, and findings should therefore be interpreted as contextually grounded accounts rather than generalisable conclusions. Social desirability bias cannot be ruled out, as participants may have emphasised professionally desirable behaviours or strict IPC adherence. The relatively small sample size further limits the representativeness of experiences and the robustness of conclusions. Finally, although the Italian to English translation was undertaken with care, specific linguistic or cultural nuances may not have been fully preserved, which may have influenced interpretative accuracy.

## Recommendations for further research

The findings highlight the need for further investigation into the effectiveness of interventions that strengthen individual, team, and organisational resilience in caring for patients with MDROs while maintaining adherence to IPC protocols. Future studies should examine how nurses adapt IPC protocols to real-world operational conditions and patient needs and identify the factors that either facilitate or constrain adaptive capacity. Organisational interventions aimed at supporting operational resilience should be evaluated for their impact on logistics, internal communication, and structural support, as well as for their ability to enable flexible care strategies without increasing emotional burden.

At the same time, IPC protocols must be designed to remain professionally and organisationally sustainable while ensuring safety standards for MDRO management. Clinical trials and multicentre studies could compare different operational configurations, degrees of procedural flexibility, and innovative decision-support tools to determine models that are feasible in routine practice. Longitudinal research would provide insight into staff well-being over time, the use and effectiveness of resilient and adaptive strategies, and trends in IPC adherence within dynamic clinical and organisational environments.

Furthermore, future work should address the limitations of Implementation Science in adapting IPC protocols to intensive care settings, where emergency-driven workflows and unstable conditions challenge the application of standardised procedures. Research should prioritise implementation approaches that acknowledge uncertainty, support real-time adaptation, and promote flexible IPC strategies suitable for high-pressure, rapidly changing clinical environments.

## Implications for policy and practice

The evidence highlights the importance of developing integrated training programmes that combine IPC competencies, decision-making in complex situations, and resilience building. Strengthening organisational structures that support nursing resilience, improving communication systems, and implementing tools for continuous monitoring of staff well-being are equally essential. IPC standardisation should be accompanied by operational flexibility, enabling nurses to adapt procedures without compromising safety. In particular, adequate staffing levels, formalised mentoring programmes, and proactive psychological support for ICU staff should be treated as structural priorities rather than optional enhancements, given their direct impact on both nursing resilience and patient safety outcomes. Healthcare organisations and implementation teams could adopt context-specific, multimodal strategies to address local barriers, make better use of underutilised interventions, and assess long-term impacts on professional practice and patient outcomes. Promoting a culture that prioritises safety and staff well-being is a fundamental prerequisite for maintaining safe, resilient, and sustainable models of care.

## Conclusion

In conclusion, this study demonstrated that nurse resilience in ICUs is a dynamic, multi-level process, intertwining individual resources, team dynamics, and organisational characteristics (S3 Fig). Adaptability, creative problem-solving, effective coordination, and situational leadership enabled nurses to balance clinical safety with IPC adherence, even under high

complexity and organisational pressure. Findings showed that nurse well-being depends not only on personal resources (self-efficacy, emotional regulation, advanced competencies) but also on professional relationships (mutual support, coordination, rapid information sharing) and a supportive organisational culture (leadership, logistics, timely communication). These three levels form the micro-foundations of organisational resilience, directly influencing clinical safety and the quality of infection prevention and control practices (S3 Table).

Simultaneously, staff face constant and often invisible pressures: the need to "always endure," ongoing emotional strain, and an increased risk of burnout, particularly when organisational culture is lacking or structural resources are insufficient. These dynamics are further compounded by a tension distinctive to ICU environments, the simultaneous management of life-threatening clinical instability and strict infection control requirements, which amplifies both the demands placed on resilience and the consequences of its failure. Critically, when resilience becomes an institutional expectation rather than a personal resource, it risks normalising structural deficiencies by displacing systemic failures onto individual practitioners. Ultimately, nursing resilience cannot compensate for structural deficiencies, however, it can be strengthened and sustained through integrated training programmes, clinical resilience models, adequate infrastructure, and stable organisational policies. In particular, adequate staffing levels, formalised mentoring programmes, and proactive psychological support should be treated as structural priorities, given their direct impact on nursing resilience, IPC adherence, and patient safety outcomes. Strengthening an organisational culture that supports staff not only promotes nurse well-being but also ensures the safe management of patients with MDROs and adherence to IPC protocols in highly critical contexts.

## Supporting information

**S1 Fig. Observational vignette 1 – ICU admission in the context of suspected MDRO infection.**
(DOCX)

**S2 Fig. Observational vignette 2 – Intra-hospital transfer of an MDRO patient to Radiology.**
(DOCX)

**S3 Fig. Thematic overview.**
(DOCX)

**S1 Table. Guide for Non-Participant Observations.**
(DOCX)

**S2 Table. Sociodemographic and professional characteristics.**
(DOCX)

**S3 Table. Themes, Categories, and Coding from Interviews.**
(DOCX)

**S4 Table. The most representative quotes.**
(DOCX)

**S1 File. File Standards for Reporting Qualitative Research (SRQR).**
(DOCX)

## Acknowledgments

I sincerely thank all the head nurses, nurses, nurse assistants, and doctors who made this research possible: Chiara Alfier, Cristina Cinquetti, Cristina Tintori, Domenico Gelormini, Elda Dani, Elena Boscolo, Elena Falcioni, Elisa Sorio, Emanuela Morandini, Federica Magaddino, Filippo Marrani, Flavio Rocca, Guido Moratello, Jacopo Rama, Marco

Guadagnini, Meniconcini Matteo, Michela Consenti, Paolo Persona, Sara Marcotto, Sara Volpato, Stefania Leoni. We would also like to thank Jessica Malzahn for reviewing the final version of this manuscript.

## Author contributions

**Conceptualization:** Eva Cappelli, Federica Canzan.

**Data curation:** Eva Cappelli.

**Formal analysis:** Eva Cappelli, Federica Canzan.

**Investigation:** Eva Cappelli.

**Methodology:** Eva Cappelli, Federica Canzan.

**Project administration:** Eva Cappelli, Marianna Azzolini, Cristina Ferrari.

**Supervision:** Federica Canzan.

**Validation:** Eva Cappelli, Federica Canzan.

**Visualization:** Eva Cappelli, Federica Canzan.

**Writing – original draft:** Eva Cappelli.

**Writing – review & editing:** Marianna Azzolini, Cristina Ferrari, Federica Canzan.

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
