## [Decision Letter · Decision Letter 0]

22 Mar 2026

Dear Dr. Cappelli,

Thank you for submitting your manuscript to PLOS ONE. After careful consideration, we feel that it has merit but does not fully meet PLOS ONE’s publication criteria as it currently stands. Therefore, we invite you to submit a revised version of the manuscript that addresses the points raised during the review process.

We look forward to receiving your revised manuscript.

Kind regards,

Eleni Magira

Academic Editor

PLOS One

**Journal Requirements:**

https://journals.plos.org/plosone/s/file?id=wjVg/PLOSOne_formatting_sample_main_body.pdf andandandand

2. In the online submission form you indicate that your data is not available for proprietary reasons and have provided a contact point for accessing this data. Please note that your current contact point is a co-author on this manuscript. According to our Data Policy, the contact point must not be an author on the manuscript and must be an institutional contact, ideally not an individual. Please revise your data statement to a non-author institutional point of contact, such as a data access or ethics committee, and send this to us via return email. Please also include contact information for the third party organization, and please include the full citation of where the data can be found.

3. Please amend your authorship list in your manuscript file to include authors names.

4. Please upload a copy of Figures 1 and 2, to which you refer in your text on page 13. If the figure is no longer to be included as part of the submission please remove all reference to it within the text.

5. If tables are embedded in the manuscript and ALSO loaded as separate files, please delete the separate files, leaving only the tables in the manuscript file.

6. We note that your paper includes detailed descriptions of individual patients/participants. As per the PLOS ONE policy (http://journals.plos.org/plosone/s/submission-guidelines#loc-human-subjects-research) on papers that include identifying, or potentially identifying, information, the individual(s) or parent(s)/guardian(s) must be informed of the terms of the PLOS open-access (CC-BY) license and provide specific permission for publication of these details under the terms of this license. Please download the Consent Form for Publication in a PLOS Journal (http://journals.plos.org/plosone/s/file?id=8ce6/plos-consent-form-english.pdf). The signed consent form should not be submitted with the manuscript, but should be securely filed in the individual's case notes. Please amend the methods section and ethics statement of the manuscript to explicitly state that the patient/participant has provided consent for publication: “The individual in this manuscript has given written informed consent (as outlined in PLOS consent form) to publish these case details”.

7. We note that there is identifying data in the Supporting Information file < Supporting Information.docx >. Due to the inclusion of these potentially identifying data, we have removed this file from your file inventory. Prior to sharing human research participant data, authors should consult with an ethics committee to ensure data are shared in accordance with participant consent and all applicable local laws.

-Location data

Please remove or anonymize all personal information, ensure that the data shared are in accordance with participant consent, and re-upload a fully anonymized data set. Please note that spreadsheet columns with personal information must be removed and not hidden as all hidden columns will appear in the published file.

**Additional Editor Comments:**

The study uses a descriptive qualitative approach, which is appropriate for exploring nurses’ lived experiences and perspectives on IPC adherence. Justification for the method is provided, emphasizing the capture of contextually grounded narratives. The authors mentioned multiple strategies for credibility (multidisciplinary team, independent analysis, coding comparisons) and the study adheres to SRQR standards, increasing transparency and methodological reliability. Several weak points are arisen:

Findings are dense and could benefit from subheadings for each resilience level (individual, team, organizational).

Transitions between participant data and theory are sometimes abrupt, making interpretation harder.

Reorganize into sub-sections: Individual resilience, Team dynamics and mutual support, Organizational and leadership factors. Consider visual aids (e.g., a diagram showing resilience across levels) to improve readability and impact.

Consider presenting raw data first, followed by interpretation in Discussion.

While stressors and negative aspects of resilience are described, the discussion could more explicitly link findings to implications for ICU management, policy, and nurse well-being.

Add in Discussion critical commentary on limits of resilience, barriers in ICU settings, and how findings align or differ from existing literature.

In Discussion, Emphasize practical implications of findings for ICU staffing, mentoring, and mental health support.

Reviewers' comments:

Reviewer's Responses to Questions

**Comments to the Author**

1. Is the manuscript technically sound, and do the data support the conclusions?

Reviewer #1: Yes

2. Has the statistical analysis been performed appropriately and rigorously?

Reviewer #1: Yes

3. Have the authors made all data underlying the findings in their manuscript fully available?

Reviewer #1: Yes

4. Is the manuscript presented in an intelligible fashion and written in standard English?

Reviewer #1: Yes

Reviewer #1: Dear Authors,

The manuscript presents a qualitative study exploring resilience among intensive care unit (ICU) nurses and how adaptive practices extend beyond infection prevention and control (IPC) protocols in the management of multidrug-resistant organisms (MDROs). The topic is relevant and of clinical importance, and the manuscript is generally written in a clear scientific style with an appropriate methodological approach to some extent. However, several points should be addressed before the manuscript can be considered for publication:

1. Although the purposive snowball sampling approach is appropriate for targeting experienced ICU professionals, it inherently carries a risk of selection bias and limited generalizability. While the clearly defined inclusion criteria may enhance the relevance and depth of the collected data, these measures do not eliminate the methodological limitations associated with non-probability sampling techniques.

The authors are therefore required to provide a clear and explicit statement of study limitations. The use of purposive snowball sampling may introduce selection bias and restrict the generalizability of the results; this should be clearly discussed in the manuscript.

2. The results section presents extensive quotations from nurses and physicians. While these insights are valuable, the current presentation is somewhat lengthy. It would be more effective to summarize key findings within the main text and consider presenting detailed quotations (e.g., from nurses, head nurses, and physicians) in a supplementary table. This would improve readability and maintain focus on the main outcomes of the study.

Please check the attached file.

.

Reviewer #1: **Yes:** Awatif Abid Al-JudaibiAwatif Abid Al-JudaibiAwatif Abid Al-JudaibiAwatif Abid Al-Judaibi

---

## [Author Response · Author response to Decision Letter 1]

4 Apr 2026

Response to Reviewers

We wish to express our sincere gratitude to the Reviewers and the Editorial Board for the thoroughness and constructiveness of their comments. Their observations have allowed us to substantially restructure and strengthen the manuscript across multiple dimensions — methodological, ethical, analytical, and practical.

The revision process has led to a more transparent and rigorous presentation of the study's design and findings, a more coherent alignment between the theoretical framework, the results, the discussion, and the conclusions, and a clearer articulation of the practical implications for ICU nursing practice, organisational policy, and infection prevention and control.

We believe that the revised manuscript now more fully reflects the standards of methodological rigour and scientific transparency expected by the journal, while remaining grounded in the lived experiences of the healthcare professionals who participated in this study.

We hope that the revised version meets the expectations of the Reviewers and the Editorial Board, and we remain available for any further clarification or amendment that may be required.

Reviewer #1 comment:

1. Although the purposive snowball sampling approach is appropriate for targeting experienced ICU professionals, it inherently carries a risk of selection bias and limited generalizability. While the clearly defined inclusion criteria may enhance the relevance and depth of the collected data, these measures do not eliminate the methodological limitations associated with non-probability sampling techniques. The authors are therefore required to provide a clear and explicit statement of study limitations. The use of purposive snowball sampling may introduce selection bias and restrict the generalizability of the results; this should be clearly discussed in the manuscript.

Authors' response:

We thank the reviewer for this important methodological observation. In accordance with the reviewer's request, we have added a clear and explicit statement regarding the limitations of purposive snowball sampling to the Limitations section of the manuscript. The added text is highlighted in blue in the revised manuscript and reads as follows: p. [27], ll. [671–675]

"The use of purposive and snowball sampling, while appropriate for this qualitative design, introduces an inherent risk of selection bias; the snowball mechanism may have disproportionately engaged nurses with higher professional visibility, potentially excluding perspectives associated with disengagement or critical views of organisational practices, and findings should therefore be interpreted as contextually grounded accounts rather than generalisable conclusions."

We believe this revised text addresses the reviewer's concern by providing a clear and explicit statement of the limitations associated with the sampling strategy, while also acknowledging that the inclusion criteria, although carefully designed, do not eliminate these inherent constraints. The revised wording has been incorporated into the Limitations section of the manuscript. We trust that this revision meets the reviewer's expectation.

2. The results section presents extensive quotations from nurses and physicians. While these insights are valuable, the current presentation is somewhat lengthy. It would be more effective to summarize key findings within the main text and consider presenting detailed quotations (e.g., from nurses, head nurses, and physicians) in a supplementary table. This would improve readability and maintain focus on the main outcomes of the study.

Authors' response:

We thank the reviewer for this constructive suggestion, which we believe has substantially improved the readability and analytical focus of the Results section. In accordance with the reviewer's recommendation, we have restructured the presentation of findings as follows:

- Reduced the number of direct quotations in the main text, retaining only the most analytically essential quotations per category, those that are most representative and/or are later discussed in the Discussion section.

- Created a new supplementary table (S3 Table) where we have relocated the supporting and illustrative quotations, organised by theme, category, participant role, and full quotation text.

- Revised the narrative structure of each subsection to present key findings in a more concise, analytical manner, with explicit references to the supplementary table where additional quotations can be found.

Below we provide a detailed account of the modifications made to each theme, with specific page and line references.

Modifications to Theme 1 — Individual Resilience and Team Dynamics

Subsection Modification Location

Self-Efficacy Retained the most representative quotation (HN2: "A newly hired nurse caring for a patient reminded the cardiac surgeon and their trainees to comply with PPE use..."); relocated the supporting quotation (N3) to Table 3. p. [21], ll. [501]

Professional Competencies Retained two key quotations: N1 (spatial adaptation: "Available spaces are often limited and cluttered… with the support nurse, we tried to redesign and keep the passage clear") and N4 (device management: "We ensured the sterile management of the external ventricular drain… using PPE each time before approaching the patient"). Relocated quotations from HN1 and N3 to Table 3. p. [21], ll. [501]

Positive Emotions Retained both quotations (N6: emotional stability; NA1: family education), as they address distinct conceptual dimensions. No relocation was necessary.

Nurse Well-being Retained three quotations as embedded fragments (N3: "constant hyper-vigilance"; N5: "infection risk, including the possibility of transmitting pathogens to their own families"; N4: "I am neither recognised nor respected by some ward physicians and external consultants"). Removed one generic fragment (HN2: "rules for proper management of infected patients") and relocated one quotation (N3: "excessive administrative burdens and exhausting shifts") to Table 3. p. [21], ll. [501]

Modifications to Theme 2 — Adaptive Strategies under Structural and Operational Constraints

Subsection Modification Location

Dynamic adaptation of spaces and protocols Retained the most synthetic quotation (N2: "We demarcated the area… limited the materials… to prevent contamination"). Relocated two supporting quotations (N7 and N3) to Table 3. p. [21], ll. [501]

Managing trade-offs between rapid decisions and safety Retained the key quotation from a physician (D4: "When patient monitor alarms sound, multiple life-saving manoeuvres are performed urgently… After the emergency, we return to order and follow protocols carefully"). Relocated the supporting operational quotation (N9) to Table 3. p. [21], ll. [501]

Cooperative communication and coordination Retained three quotations: N4 (peer support: "We support new staff facing unwelcoming dynamics from long-standing physicians"), N3 (head nurse role: "She contacted the Emergency Department head nurse to understand why handovers were missing") and “We need more structured briefings with physicians to manage patients and critical events…” [N2].Relocated two supporting quotations (N5 and N4) to Table 3. p. [21], ll. [501]

Modifications to Theme 3 — Organisational and Leadership Factors

Subsection Modification Location

Available resources and structured IPC processes Retained the key quotation on procedural flexibility (N2: "The substance doesn't change, but the approach does… when unpredictable, re-engineering is necessary"). Relocated two supporting quotations from N3 (poorly functioning lifts; gaps in interdepartmental communication) to Table 3. p. [21], ll. [501]

Organisational culture of infection risk Retained three quotations: N2 (inter-institutional discontinuity: "The same type of patient is managed differently in the ICU and rehabilitation"); D2 (nurses as "guardians of the rules") embedded; HN1 (broader organisational challenges: "Heterogeneous protocols between hospitals and rehabilitation facilities, incomplete or delayed communications..."). Removed one incomplete fragment (N1: "Despite clear signs and instructions"). p. [21], ll. [501]

All quotations relocated from the main text have been compiled into S3 Table, structured as follows:

Theme Category Participant Quotation

Theme 1. Individual and Team Resilience of Nurses

Self-Efficacy: personal efficacy and situational initiative N3 “I managed a patient transfer to the MRI; it was very challenging with many devices, monitors, and limited space… I tried to ensure the best adherence to PPE use”.

Theme 1. Individual and Team Resilience of Nurses

Professional Competencies: advanced technical skills HN1 “I reorganised the nurse’s workload, temporarily assigning another patient to a nearby colleague”.

Theme 1. Individual and Team Resilience of Nurses Professional Competencies: advanced technical skills N3 “We always update all device-maintenance information promptly, which is often more challenging in the ICU than in other wards”.

Theme 1. Individual and Team Resilience of Nurses

Nurse Well-being: stress factors

N3 “Excessive administrative burdens and exhausting shifts”

Theme 2. Nurse Adaptive Strategies in the ICU Dynamic adaptation of spaces and protocols

N7 “We used screens to create a ‘bubble’ reminding staff to wear PPE before entering”.

Theme 2. Nurse Adaptive Strategies in the ICU Dynamic adaptation of spaces and protocols N3 “Hand sanitiser dispensers were placed strategically… near the service table and monitors”.

Theme 2. Nurse Adaptive Strategies in the ICU

Managing trade-offs between rapid decisions and safety N9 "One of us stays outside the room to pass all necessary items without risking contamination".

Theme 2. Nurse Adaptive Strategies in the ICU Cooperative communication and coordination N5 "Or I remind my distracted colleague who forgot to wear the gown correctly".

Theme 2. Nurse Adaptive Strategies in the ICU Cooperative communication and coordination N4 "Promptly calling Radiology to confirm the MRI booking for an obese, infected patient".

Theme 3. Interaction between Nurse Resilience and Organisational Support Available resources and structured IPC processes

N3 "Poorly functioning lifts designated for infected patients".

Theme 3. Interaction between Nurse Resilience and Organisational Support Available resources and structured IPC processes

N3 "Gaps in interdepartmental communication and incomplete paper documentation”.

Theme 3. Interaction between Nurse Resilience and Organisational Support Organisational culture of infection risk N1 "Despite clear signs and instructions”.

We believe these revisions have improved the clarity and analytical focus of the Results section, while preserving the richness of participant voices through the supplementary table. We thank the reviewer again for this valuable suggestion and trust that the revised version now meets the journal's standards for presentation.

Journal Requirements comment:

Authors' response:

We thank the Editorial Office for this reminder. We have carefully reviewed and revised the manuscript to ensure full compliance with PLOS ONE's style and formatting requirements, following the guidelines provided in the official formatting sample templates for both the main body and the title/authors/affiliations sections. All relevant adjustments have been implemented accordingly in the revised version of the manuscript.

2. In the online submission form you indicate that your data is not available for proprietary reasons and have provided a contact point for accessing this data. Please note that your current contact point is a co-author on this manuscript. According to our Data Policy, the contact point must not be an author on the manuscript and must be an institutional contact, ideally not an individual. Please revise your data statement to a non-author institutional point of contact, such as a data access or ethics committee, and send this to us via return email. Please also include contact information for the third-party organization, and please include the full citation of where the data can be found.

Authors' response:

We thank the Editorial Office for this important observation. In accordance with PLOS ONE's Data Policy, we have revised the Data Availability Statement. The contact point has been changed from the corresponding author to the South-West Veneto Area Territorial Ethics Committee (Comitato Etico Territoriale Area Vasta Sud-Ovest Veneto, Prot. No. 68603), which is the institutional body that approved this study and is independent from the authorship team. The updated contact information for the Ethics Committee is provided in the revised title page, attached herewith. We apologise for the oversight and confirm that the revised title page has been updated accordingly.

3. Please amend your authorship list in your manuscript file to include authors names.

Authors' response:

We thank the editorial office for this observation. We confirm that the authorship list has been reviewed and amended in the revised manuscript file. All authors' full names are now clearly included in the title page: Eva Cappelli, Marianna Azzolini, Cristina Ferrari, and Federica Canzan, with their respective affiliations and contribution statements.

4. Please upload a copy of Figures 1 and 2, to which you refer in your text on page 13. If the figure is no longer to be included as part of the submission, please remove all reference to it within the text.

Authors' response:

We thank the Editorial Office for raising this point. We would like to clarify that both Figure 1 and Figure 2 were included in the original submission. If they were not visible, this may have been due to a technical issue during the upload process. We apologise for any inconvenience this may have caused.

In the revised submission, both figures have been embedded within the manuscript text at the appropriate locations, as per the journal's guidelines. We trust that the figures are now correctly accessible and hope this resolves the issue.

5. If tables are embedded in the manuscript and ALSO loaded as separate files, please delete the separate files, leaving only the tables in the manuscript file.

Authors' response:

We thank the Editorial Office for this observation. We confirm that the separately uploaded table files have been removed. All tables are now exclusively embedded within the manuscript file.

6. We note that your paper includes detailed descriptions of individual patients/participants. As per the PLOS ONE policy (http://journals.plos.org/plosone/s/submission-guidelines#loc-human-subjects-research) on papers that include identifying, or potentially identifying, information, the individual(s) or parent(s)/guardian(s) must be informed of the terms of the PLOS open-access (CC-BY) license and provide specific permission for publication of these details under the terms of this license. Please download the Consent Form for Publication in a PLOS Journal (http://journals.plos.org/plosone/s/file?id=8ce6/plos-consent-form-english.pdf). The signed consent form should not be submitted with the manuscript but should be securely filed in the individual's case notes. Please amend the methods section and ethics statement of the manuscript to explicitly state that the patient/participant has provided consent for publication: “The individual in this manuscript has given written informed consent (as outlined in PLOS consent form) to publish these case details”.

Authors' response:

The authors thank the Editorial Office for raising this important ethical issue, which has been addressed with the utmost seriousness. We would first like to clarify the methodological nature of the content originally presented as "case studies". The study was d

---

## [Decision Letter · Decision Letter 1]

13 Apr 2026

Resilient nursing in ICU: adaptive practices beyond IPC protocols for MDRO management. A Qualitative Study

PONE-D-26-04881R1

Dear Dr. Eva Cappelli,

We’re pleased to inform you that your manuscript has been judged scientifically suitable for publication and will be formally accepted for publication once it meets all outstanding technical requirements.Please note that  there are two titles for the tables: the original title appears above the table, and the revised title is placed below it. Please remove the original title and place the revised title above the table.

Kind regards,

Eleni Magira

Academic Editor

PLOS One

---

## [Editor Report · Acceptance letter]

PONE-D-26-04881R1

PLOS One

Dear Dr. Cappelli,

I'm pleased to inform you that your manuscript has been deemed suitable for publication in PLOS One. Congratulations! Your manuscript is now being handed over to our production team.

Kind regards,

on behalf of

Dr. Eleni Magira

Academic Editor

PLOS One